# Assessment of the Severity of Intermediate Coronary Artery Stenosis Using the Systemic Inflammatory Response Index

**DOI:** 10.3390/diagnostics15020162

**Published:** 2025-01-13

**Authors:** Serdar Akyel, Ahmet Korkmaz, Abdülkadir Yıldız

**Affiliations:** 1Department of Cardiology, School of Medicine, Kastamonu University, Kastamonu 37150, Turkey; gadirr@yahoo.com; 2Department of Cardiology, Ankara City Hospital, Ankara 06200, Turkey; drahmtkrkmz07@gmail.com

**Keywords:** systemic inflammatory response index (SIRI), fractional flow reserve (FFR), coronary artery disease

## Abstract

**Background**: Fractional Flow Reserve (FFR) is a method that enables the hemodynamic assessment of coronary artery stenosis. The Systemic Inflammatory Response Index (SIRI) is a new marker calculated by multiplying the neutrophil-to-lymphocyte ratio (NLR) with the monocyte count. It is indicative of the presence and severity of coronary artery disease. This study evaluates the relationship between the functional significance of FFR measurements and the SIRI in intermediate coronary stenosis. **Methods**: A total of 294 patients with 50–70% stenosis in their coronary arteries based on quantitative measurement following angiography who underwent FFR measurement were included in the study before the FFR procedure. Total and differential leukocyte counts and routine biochemical tests were performed. **Results**: A total of 37% of the patients were found to have a positive FFR, while 63% had a negative FFR. Significant differences were observed in the neutrophil count, monocyte count, Systemic Inflammation Response Index (SIRI), total cholesterol, and amount of adenosine used between the groups (*p* < 0.05). A SIRI value of 1.16 was 77% sensitive and 55% specific for FFR positivity. Multivariate logistic regression analysis identified the SIRI as an independent predictor of FFR positivity. **Conclusions**: Our study has demonstrated that high values of the SIRI may serve as a new biomarker for predicting FFR positivity.

## 1. Introduction

Ischemic heart diseases remain the primary cause of mortality and disability-adjusted life-year loss [1]. Today, coronary artery disease is the most important cause of mortality and morbidity in the world and our country. At the beginning of the 21st century, 50% of all deaths in developing countries were due to cardiovascular diseases (CVD), while this rate was 25% in developed countries. According to current knowledge, atherosclerosis is a multifactorial process in which chronic inflammation plays a role at every stage in its progression, from the beginning to the later period, and in which many risk factors contribute to the developing inflammation. Atherosclerosis is a systemic arterial disease that affects vessels of all sizes, primarily large and medium-sized elastic arteries. Atherosclerosis is a process that occurs in the form of intimal smooth muscle cell accumulation and proliferation; the infiltration of monocytes, macrophages, and T lymphocytes; a connective tissue matrix rich in collagen, elastin fibers, fibronectin, and proteoglycans; and lipid storage, especially in the form of free cholesterol or cholesterol esters within the cells and in the surrounding connective tissue.

Many methods have been used to diagnose coronary artery disease from the past to the present. Coronary angiography is the fundamental method used in the detection and intervention of coronary artery diseases [2]. As a result of coronary angiography, a qualitative evaluation of coronary lesions and stenosis is made. Qualitative assessment of coronary stenosis post-coronary angiography is only sometimes reliable. Moreover, an anatomically significant lesion may not be hemodynamically significant. Fractional Flow Reserve (FFR) is a method that enables the hemodynamic assessment of coronary artery stenoses. It is particularly beneficial in the hemodynamic evaluation of intermediate stenoses between 50 and 60% [3].

Studies have identified that endothelial dysfunction, oxidative stress, platelet activation, and inflammation play crucial roles in the development and progression of atherosclerosis [4,5]. Studies have shown that laboratory levels of the neutrophil–lymphocyte ratio (NLR) and platelet–lymphocyte ratio (PLR) indicate inflammation. They can provide insights into the severity of coronary artery disease. In addition, the NLR has been found helpful in predicting in-hospital mortality in patients presenting with acute decompensation of atrial fibrillation [6]. Recently, the Systemic Inflammation Response Index (SIRI, calculated by multiplying the NLR with the monocyte count), a newly introduced marker, has been shown to indicate the presence and severity of coronary artery disease [7]. The hemogram result, a simple test, can inform us about whether one has coronary artery disease. This study evaluates the relationship between the functional significance of FFR measurements and the SIRI in intermediate coronary stenoses.

## 2. Materials and Methods

This retrospective study reviewed records from two centers between 1 January 2012 and 31 December 2022. A total of 339 patients who had 50–70% stenosis in their coronary arteries based on quantitative measurement following angiography and who underwent FFR measurement were included in the study. The patients were included in the survey, respectively, regardless of age, gender, and ethnicity. The patients considered for the study were those with stable coronary artery disease undergoing coronary angiography. Patients with acute coronary syndrome, moderate or severe valvular heart disease, significant arrhythmia, hemodynamic instability, secondary stenosis in the assessed coronary artery, more than 40% stenosis in other coronary arteries, previous coronary artery bypass surgery or percutaneous coronary intervention, acute–chronic or inflammatory bowel disease, anemia, chronic kidney failure, and malignancy were excluded from the study (Table 1). When patients were separated from the study according to the exclusion criteria, the remaining 294 patients were included. (Figure 1). All the patients meet the inclusion criteria.

Hospital records were utilized to ascertain patients’ angiographic information, clinical data, and demographic characteristics. Blood samples, including lipid profiles, serum creatinine values, and complete blood counts, were taken upon hospital admission. Hemogram tests of both FFR (+) and FFR (−) groups were measured automatically in the XN 1000 (Sysmex, Kobe, Japan). SIRI values were calculated by multiplying the NLR with the monocyte count.

For coronary narrowing ranging from 50% to 70%, a decision for Fractional Flow Reserve (FFR) measurement was made at the cardiologist’s discretion. After administering 5000 units of intracoronary heparin, the coronary artery was evaluated using a non-side-hole guiding catheter. Following calibration, a 0.014-inch pressure-monitoring guide wire (Prime Wire, Volcano, San Diego, CA, USA) was positioned distal to the lesion. Before FFR measurement, a 200 mcg intracoronary bolus of nitroglycerin was administered. Distal intracoronary measurements were taken initially and after achieving hyperemia with incrementally increased doses of adenosine. The ratio of the mean intracoronary pressure at peak hyperemia to the mean aortic pressure determined the FFR value.

An FFR value ≤ 0.80 was considered functionally significant. Patients with FFR values > 0.80 were classified as Group 1, and those with FFR values ≤ 0.80 as Group 2.

The study protocol was approved by the Kastamonu University Education and Research Hospital Ethics Committee and was performed in accordance with the Declaration of Helsinki. All patients provided written informed consent to participate in the study.

## 3. Results

The study included 294 patients, comprising 79 women (27%) and 215 men (73%). A total of 37% of the patients were found to have a positive FFR, while 63% had a negative FFR (Figure 2). The essential clinical and demographic characteristics, summarized in Table 2, showed no significant differences in age and gender between the groups with significant and non-significant FFRs (*p* > 0.05). Additionally, no significant differences were observed in the fasting blood sugar levels, creatinine levels, presence of diabetes and hypertension, and smoking habits between the two groups (*p* > 0.05).

Significant differences were observed in the neutrophil count, monocyte count, Systemic Inflammation Response Index (SIRI), total cholesterol, and amount of adenosine used between the groups (*p* < 0.05). The group with a non-significant FFR showed a lower neutrophil count (4.62 [3.69–5.68] vs. 5.32 [3.80–6.50]; *p*: 0.003), monocyte count (0.56 [0.5–0.7] vs. 0.65 [0.56–0.78]; *p* < 0.001), SIRI (1.10 [0.87–1.54] vs. 1.56 [1.18–2.13]; *p* < 0.001), and total cholesterol (185 [151–214] vs. 204 [162–234]; *p* = 0.008), while the amount of adenosine used was less (250 [150–300] vs. 200 [104–250]; *p*: 0.003) in the group with a significant FFR (Table 3).

The ROC analysis revealed an area under the curve (AUC) of 0.686 CI (0.622–0.750) (*p* < 0.001). A SIRI value of 1.25 was 71% sensitive and 61% specific for FFR positivity (Figure 3).

Multivariate logistic regression analysis identified the SIRI as an independent predictor of FFR positivity. (Odds ratio: 2.805; 95% CI [1.877–4.192]; *p* < 0.001) (Table 4).

## 4. Discussion

This study found that coronary narrowing was deemed functionally significant after Fractional Flow Reserve (FFR) measurements were independently associated with the Systemic Inflammation Response Index (SIRI) levels.

Oxidative stress and inflammation are fundamental mechanisms in the development and progression of atherosclerosis [4,5,8]. In coronary artery disease (CAD), standard inflammatory markers, such as white blood cells (WBCs) and C-reactive protein (CRP), have been linked with high cardiovascular risk [9], the severity of coronary artery disease [10], the instability of atherosclerotic plaque [11], and mortality related to CAD [12]. Chronic low-grade inflammation is believed to play a critical role in the progression of coronary artery disease [13]. Leukocytes and their subtypes (neutrophils, lymphocytes, and monocytes) can be measured in the total blood cell count, offering an economical and accessible method to assess inflammatory processes involved in the pathogenesis of CAD [14]. Monocytes, a specific subtype of lymphocytes, contribute to the initiation and progression of atherosclerosis by releasing pro-inflammatory cytokines, free oxygen radicals, and proteolytic enzymes [15]. They adhere to the endothelium, transform into macrophages, and become foam cells by absorbing lipid particles, activating cytokines and free oxygen radicals [16]. Studies have identified monocyte count as an independent short- and long-term marker of cardiovascular disease-related mortality, irrespective of other risk factors [17,18]. Neutrophils, the most common leukocyte subtype, exacerbate vascular wall inflammation by inducing small muscle cell apoptosis [19,20]. High neutrophil counts are positively associated with plaque rupture risk [20] and increase the risk of thrombosis in the microcirculation [21]. Conversely, lymphocytes inhibit the progression of atherosclerosis [22]. Lymphopenia is positively associated with Major Adverse Cardiac Events (MACE) [23] and is linked with an increased prevalence of heart failure [24] and poor prognosis in patients with acute coronary syndrome (ACS) [25]. Platelets influence atherosclerosis in two ways: their adhesion to the vessel wall facilitates plaque development [26], while their activation triggers inflammation and clot formation [27].

Recently, some parameters obtained by dividing white blood cell counts have been revealed due to their success in showing cardiovascular risk, myocardial infarcts, and the prevalence of atherosclerosis due to the results that can be easily calculated from hemogram results [28,29].

The monocyte–lymphocyte ratio (MLR), neutrophil–lymphocyte ratio (NLR), and platelet–lymphocyte ratio (PLR) serve as biomarkers reflecting a variety of immune pathways and cell functions. By integrating the effects of two distinct cell types that interact, their predictive value for cardiovascular disease (CVD) and mortality is enhanced synergistically [30,31,32].

The monocyte–lymphocyte ratio (MLR) is a parameter obtained by dividing monocytes by lymphocytes. In a study by Hua Yang et al., a higher baseline monocyte–lymphocyte ratio (MLR) was linked to an increased risk of death among US adults. MLR proved to be a strong independent predictor of overall mortality and cardiovascular mortality in the general population [33]. Similarly, research by Fan et al. found a significant association between an elevated MLR and a greater risk of mortality in the six months following intervention in patients with acute myocardial infarction (AMI) who underwent percutaneous coronary intervention (PCI) [34].

Another marker is the neutrophil–lymphocyte ratio (NLR), calculated by dividing the number of neutrophils by the number of lymphocytes. The NLR is the most researched white blood cell ratio related to cardiovascular diseases. Nunez et al. found that patients diagnosed with ST-elevation myocardial infarction (STEMI) had the highest neutrophil–lymphocyte ratio (NLR) within 12 to 24 h after being admitted to the hospital [35]. Various studies have shown a link between an elevated NLR and an increased occurrence of no-reflow following percutaneous coronary intervention (PCI) in STEMI patients. Furthermore, long-term follow-up of non-ST elevation myocardial infarction (NSTEMI) patients with slow coronary flow seen on angiography and a high NLR (greater than 3.88) was independently associated with a higher rate of recurrent myocardial infarction. A significant relationship was also observed between the NLR and myocardial injury, as reflected by elevated CK-MB levels. The NLR also demonstrated a negative correlation with cardiac contractility. Thus, a high NLR serves as a strong indicator of the extent of myocardial damage, suggesting that the level of inflammation is associated with compromised contractile function in the heart [36].

Another marker introduced is the platelet–lymphocyte ratio (PLR), obtained by dividing the platelet count by the lymphocyte count. As a cost-effective and widely accessible biomarker, the platelet–lymphocyte ratio (PLR) incorporates both inflammatory and thrombotic pathways, potentially offering more excellent prognostic value than individual platelet or lymphocyte counts. This makes it a promising tool for predicting acute coronary syndrome (ACS) [37,38]. Ye et al. found that a higher PLR was linked to worse clinical outcomes in patients with acute heart failure (HF), suggesting it could serve as a novel marker in the management of acute HF [39]. Additionally, a recent study by Turcato et al. demonstrated that PLR was independently associated with a threefold increase in the 30-day mortality risk following emergency department admission for acute decompensated HF [40].

As far as we know, our study is the first clinical investigation to demonstrate the relationship between clinically significant narrowing detected using Fractional Flow Reserve (FFR) measurements and the Systemic Inflammation Response Index (SIRI). Previous studies have explored the relationship between the Systemic inflammation Index (SII) and the extent of coronary artery disease. For instance, in their study involving 400 patients undergoing coronary angiography, Liu and colleagues investigated the relationship between the Gensini score, indicating the severity of coronary artery disease, and the SII. They concluded that the SII is an independent risk factor for diagnosing and determining the severity of coronary artery disease [7]. Similarly, a study by Candemir et al. involving 669 patients found a positive correlation between coronary artery severity measured using the SYNTAX score and the SII [41]. Additionally, Erdoğan et al. discovered a positive correlation between the SII and the likelihood of detecting functionally significant narrowing in the coronary artery using FFR [42].

One of the critical differences between the SIRI and SII is that the SIRI incorporates the monocyte count, whereas the SII includes the platelet count. This could contribute significantly to the results, as monocytes are the primary cells responsible for atherosclerotic plaque formation. Previous studies in patients with acute coronary syndrome have observed increased inflammatory markers. The SII has been found to predict the occurrence of Major Adverse Cardiac Events (MACE) in acute coronary syndrome patients more accurately than classical risk factors [43]. These findings are supported by Li et al., who discovered the association of lymphocyte-based inflammatory markers with MACE and reported the advantage of the SIRI over other inflammatory markers in this context [44]. In another observational study spanning ten years with 85,000 participants, a correlation was found between a high SIRI and increased incidence of ACS in patients under 60. In contrast, no correlation was found with the SII [45].

The limitations of our study include a small patient cohort and relatively broad exclusion criteria, which encompassed active neoplastic processes or paraneoplastic syndromes, diagnosed active viral or bacterial infections, chronic kidney disease (stages III-V), an elevated erythrocyte sedimentation rate, and serum CRP levels. Furthermore, this study’s retrospective, cross-sectional, and observational nature limits the ability to analyze causal relationships among variables. It diminishes the confidence in attributing low-grade inflammation directly to heart disease. Due to the retrospective character of the study, other inflammatory parameters such as the ESR, fibrinogen, quantitative CRP, homocysteine, and uric acid could not be obtained in all patients, so the effects of these parameters on the NLR and SIRI could not be studied. Since the height and weight measurements of all patients included in the study were not recorded in the system, a BMI calculation could not be made. Therefore, the obesity paradox could not be ruled out. Additionally, the assessment of coronary artery disease severity was conducted using coronary angiography without accounting for the stabilizing effects of coronary calcifications. Lastly, the influence of hypolipidemic medications could not be considered, even though they may affect low-grade inflammation, as all patients received comparable doses of statins.

## 5. Conclusions

Given that the SIRI, a novel inflammatory biomarker, shows promise as a clinical tool for assessing coronary artery disease and its potential complications, further in-depth and well-designed studies on larger patient groups are necessary.

## Figures and Tables

**Figure 1 diagnostics-15-00162-f001:**
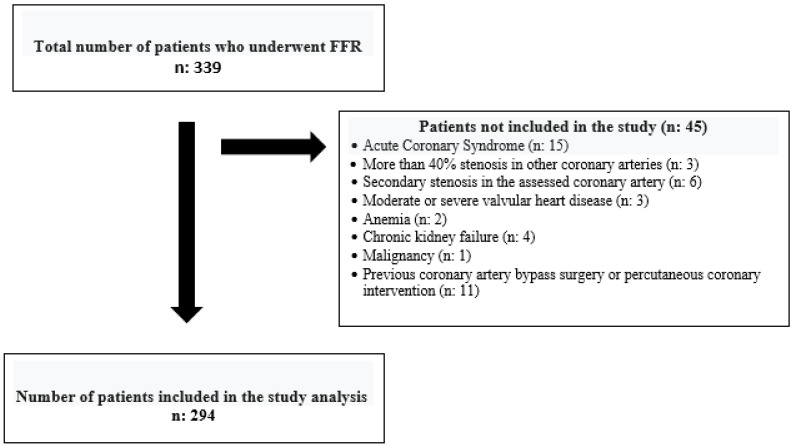
Number of patients included and excluded in the study.

**Figure 2 diagnostics-15-00162-f002:**
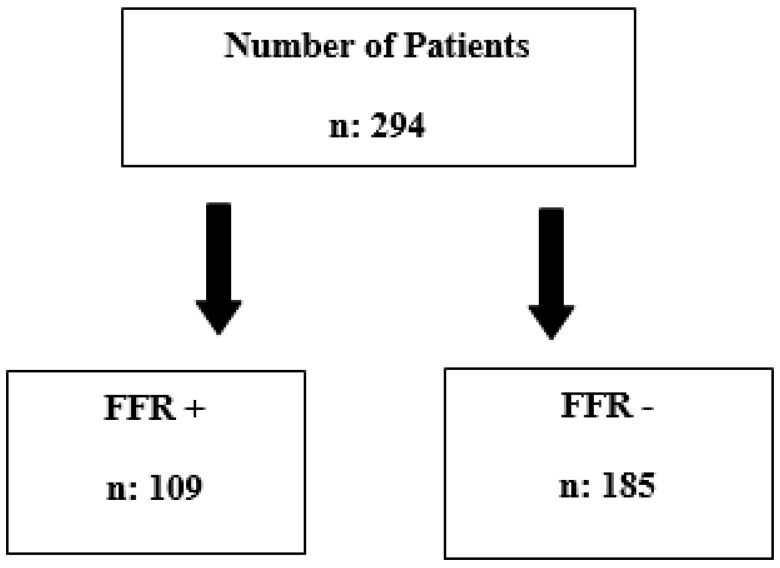
The total number of patients included in the study and grouping according to the FFR results.

**Figure 3 diagnostics-15-00162-f003:**
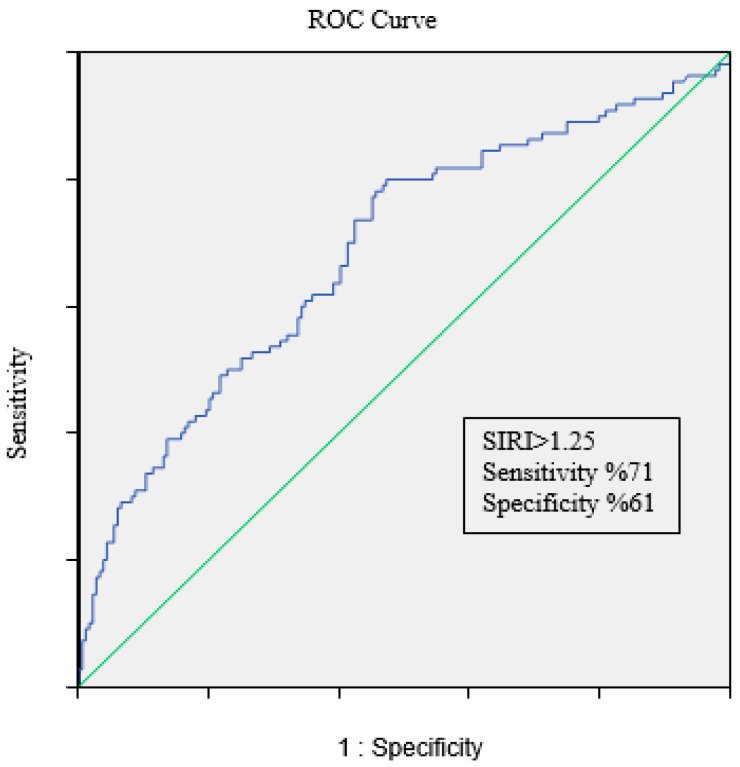
Receiver operating characteristic curve of Systemic Inflammatory Response Index (SIRI) for predicting FFR positivity.

**Table 1 diagnostics-15-00162-t001:** Study exclusion criteria.

Patients with acute coronary syndrome Moderate or severe valvular heart disease Significant arrhythmia Hemodynamic instability Secondary stenosis in the assessed coronary artery More than 40% stenosis in other coronary arteries Previous coronary artery bypass surgery or percutaneous coronary intervention Acute–chronic or inflammatory bowel disease Anemia Chronic kidney failure Malignancy

**Table 2 diagnostics-15-00162-t002:** The baseline characteristics of the study population according to the FFR significance.

	FFR Non-Significant	FFR Significant	*p* Value
Age	61 (54–70)	59 (51–66)	0.054
Male gender	128 (70%)	87 (79%)	0.079
DM2	57 (31%)	26 (24%)	0.227
HT	72 (39%)	40 (37%)	0.709
Smoking	94 (51%)	48 (44%)	0.277

Categorical variables are expressed as numbers and percentages; numerical variables are expressed as the standard deviation or median (min–max). Abbreviations: DM2: Type 2 Diabetes Mellitus; HT: hypertension.

**Table 3 diagnostics-15-00162-t003:** The baseline biochemical characteristics of the study population according to the FFR significance.

	FFR Non-Significant	FFR Significant	*p* Value
Glucose	108 (94–151)	107 (91–135)	0.719
Urea	34 (27–40)	33 (26–39)	0.635
Creatinine	0.89 (0.77–1.03)	0.93 (0.78–1.04)	0.249
eGFR	82 (63–101)	79 (64–95)	0.234
Hemoglobin	14.1 (13–15.1)	14.5 (13.4–15.4)	0.057
RDW	13.7 (13–14.6)	13.5 (13.1–14.1)	0.153
MPV	8.6 (8.1–9.5)	8.7 (8–9.3)	0.780
PCT	0.21 (0.17–0.24)	0.21 (0.16–0.23)	0.108
PDW	16.9 (16.3–18)	16.8 (16.2–17.3)	0.140
PLT	238 (207–275)	242 (197–293)	0.549
WBC	7.81 (6.73–9.30)	8.20 (6.79–9.7)	0.182
Neutrophils	4.62 (3.69–5.68)	5.32 (3.80–6.50)	0.003 *
Lymphocytes	2.26 (1.75–2.73)	2.13 (1.75–2.63)	0.218
Monocytes	0.56 (0.5–0.7)	0.65 (0.56–0.78)	<0.001 *
SII	477 (328–753)	575 (390–882)	0.009
SIRI	1.10 (0.87–1.54)	1.56 (1.18–2.13)	<0.001 *
Total cholesterol	185 (151–214)	204 (162–234)	0.008
LDL	111 (86–143)	122 (96–143)	0.068
HDL	41 (35–49)	41 (34–46)	0.142
Triglycerides	127 (93–178)	146 (104–244)	0.018
TG/HDLc	4.15 (0.81–7.49)	4.80 (1.24–8.36)	0.198
Initial FFR	0.94 (0.92–0.95)	0.88 (0.83–0.92)	<0.001 *
Final FFR	0.88 (0.84–0.91)	0.74 (0.71–0.78)	<0.001 *
Amount of adenosine	250 (150–300)	200 (104–250)	0.003 *

Categorical variables are expressed as numbers and percentages; numerical variables are expressed as the standard deviation or median (min–max). * *p* < 0.05 is considered statistically significant. Abbreviations: eGFR: Estimated Glomerular Filtration Rate; FFR: Fractional Flow Reserve; HDL: High-Density Lipoprotein; LDL: Low-Density Lipoprotein; MPV: Mean Platelet Volume; PCT: Procalcitonin; PDW: Platelet Cell Distribution Width; PLT: platelet count; RDW: Red Cell Distribution Width; SII: Systemic Inflammatory Index; SIRI: Systemic Inflammatory Response Index; TG/HDLc: Triglycerides/High-Density Lipoprotein ratio; WBC: white blood cell.

**Table 4 diagnostics-15-00162-t004:** Independent predictors of FFR positivity.

	Odds Ratio	95% Confidence Interval	*p* Value
SIRI	2.805	1.877–4.192	<0.001 *
Male gender	0.689	0.353–1.343	0.274
Age	0.984	0.958–1.010	0.232
Triglyceride	1.002	0.999–1.005	0.223
Total cholesterol	1.006	0.999–1.012	0.075
Hemoglobin	0.979	0.874–1.096	0.708

Categorical variables are expressed as numbers and percentages; numerical variables are expressed as the standard deviation or median (min–max). * *p* < 0.05 was considered statistically significant. Abbreviations: FFR: Fractional Flow Reserve; SIRI: Systemic Inflammatory Response Index.

## Data Availability

The original contributions presented in the study are included in the article; further inquiries can be directed to the corresponding author.

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
