# Peer review of "Assessment of the Severity of Intermediate Coronary Artery Stenosis Using the Systemic Inflammatory Response Index"

_diagnostics, 2025, doi:10.3390/diagnostics15020162_

Round 1
Reviewer 1 Report
Comments and Suggestions for Authors
Introduction:
- this part is too long and presents data that are not related with systemic inflammatory response index(SIRI). The authors must re-write this part. Data presented in this part must be related to the neutrophile/lymphocyte ratio, monocyte count and SIRI. An interesting paper that is related to NLR subject is the following Kundnani et all, Use of neutrophil to lymphocyte ratio to predict in-hospital mortality in patients admitted with acute decompensation of atrial fibrillation,
- resume this part at a maximum of one page.
Materials and methods:
- please mention if you analysed consecutive patients and if there were discrimination criteria related to age, gender, or ethnicity. Describe in detail the protocol used to recruit patients from these two hospitals. Add a flowchart for a more comprehensive reading.
- please mention if the complete blood count was made automatically (in this case please specify if the devices used are similar) or if this was made manually.
- in my opinion, your group 2 is almost double that of group 1 and that difference in subject number may introduce biases in the results favour of the larger group
In results:
- please add data regarding other inflammation markers such as ESR, fibrinogen, quantitative CRP, homocysteine, and uric acid, and, in the results section, analyze the impact of those parameters on the NLR and the SIRI
- please add echocardiographic data: systolic and diastolic function, tissular Doppler imaging, LV myocardial strain ...
- please note DM type, and add eGFR for your patients because at this age normal range creatinine is responsible for a decreased eGFR, and because at stage I CKD you have kidney dimension. These data are mandatory to understand that you really excluded CKD.
- introduce more confounders into your regression model. Please show us the impact of inflammation evaluated by other parameters mentioned above.
- reanalyse your ROC because you show us a low sensitivity and specificity
- use in your lipidic analysis the values of oxLDL, TG/HDLc ratio
In discussion:
- lines 196-198 are not really sustained by your results
- rearrange lines 199-265 as an introduction, those lines are not debating with your results
- analyse those studies:
10.23736/S2724-5683.20.05354-2
https://doi.org/10.2147/JIR.S421491
https://doi.org/10.3390/ijms23179553
https://cardiab.biomedcentral.com/articles/10.1186/s12933-024-02272-5#Tab6:~:text=DOI-,https%3A//doi.org/10.1186/s12933%2D024%2D02272%2D5,-Share%20this%20article
Author Response
Comments 1: this part is too long and presents data that are not related with systemic inflammatory response index(SIRI). The authors must re-write this part. Data presented in this part must be related to the neutrophile/lymphocyte ratio, monocyte count and SIRI. An interesting paper that is related to NLR subject is the following Kundnani et all, Use of neutrophil to lymphocyte ratio to predict in-hospital mortality in patients admitted with acute decompensation of atrial fibrillation,
Response 1:Dear reviewer, first of all, thank you for taking the time to evaluate our work.Within the scope of your suggestions, the introduction section was simplified and the work you gave as an example was mentioned.(page 2- lines 55-57)
Comments 2:please mention if you analysed consecutive patients and if there were discrimination criteria related to age, gender, or ethnicity. Describe in detail the protocol used to recruit patients from these two hospitals. Add a flowchart for a more comprehensive reading.Response 2 Since our study is retrospective, it was conducted on patients who underwent coronary angiography and were given an FFR decision between 2012 and 2022. All patients who did not meet the exclusion criteria, regardless of age, gender, and ethnicity, were included respectively. (page 2- lines 67-68)
Comment 3:please mention if the complete blood count was made automatically (in this case please specify if the devices used are similar) or if this was made manually. Response 3: Hemogram tests of both FFR(+) and FFR(-) groups were measured automatically in XN 1000 (Sysmex, Kobe,Japan).Since the same devices were used in both hospitals, data from both centers were used and included in the study.(page 2- lines 79-80)
Comment 4:in my opinion, your group 2 is almost double that of group 1 and that difference in subject number may introduce biases in the results favour of the larger group
Response 4:Moderate stenosis is not common in angiograms. Whether these stenoses are functionally significant is evaluated by FFR examination. For this reason, unfortunately, our number of patients is low. If only our number of patients had been larger, we could have obtained better results. If you wish we can equalise number of the groups by omiting excess patients.
Comment 5:please add data regarding other inflammation markers such as ESR, fibrinogen, quantitative CRP, homocysteine, and uric acid, and, in the results section, analyze the impact of those parameters on the NLR and the SIRI
Resonse 5:Due to the retrospective character of the study and because these patients were elective coronary angiography patients, sediment, CRP, fibrinogen, uric acid and homocysteine ​​were not routinely checked. Therefore, not all patients were included in the study as they did not have this data. As you stated, more valuable results could have been obtained if this data had been available.
Comment 6:please add echocardiographic data: systolic and diastolic function, tissular Doppler imaging, LV myocardial strain ..Response 6:Due to the retrospective character of the study and since these patients are elective coronary angiography patients, systolic and diastolic function, tissular Doppler imaging, and LV myocardial strain are not routinely examined. As you stated, more valuable results could have been obtained if this data had been available.
Comment 7:please note DM type, and add eGFR for your patients because at this age normal range creatinine is responsible for a decreased eGFR, and because at stage I CKD you have kidney dimension. These data are mandatory to understand that you really excluded CKD. Response 7:As you menbtioned eGFR values are given in the table.(Page 4, Table 2). All the patients were diabetes type 2.
Comment 8: introduce more confounders into your regression model. Please show us the impact of inflammation evaluated by other parameters mentioned above.Response 8: Unfortunately not all the patients had the other inflammatuar markers because of retrospective desing of the study which is the most important limitation of our paper.
Comment 9:reanalyse your ROC because you show us a low sensitivity and specificityResponse 9:As you mentioned we reanayse ROC analysis and change the cut off level with higher specifity. A SIRI value 1.25 was 71% sensitive and 61% specific for FFR positivity. (Page 4-Line 123)
Comment 10:use in your lipidic analysis the values of oxLDL, TG/HDLc ratioResponse 10: As you mentioned, we analyze the TG/HDLc ratio and add it to Table 2. (Page 4, Table 2). ox LDL measurement was not avaliable in our clinic..
Comment 11:lines 196-198 are not really sustained by your resultsResponse 11: In our study. Multivariate logistic regression analysis identified SIRI as an independent predictor of FFR positivity. (Odds ratio: 2.805; 95% CI [1.877-4.192]; p<0.001). (page 5-line 128)
Comment 12: rearrange lines 199-265 as an introduction, those lines are not debating with your resultsResponse 12: As SIRI includes both NLR and Monocyte count, we discussed these topics in the discussion section. I think that specifying the characteristics of these subgroups makes the results easier for the reader to understand so that the leukocyte subtypes and inflammatory markers mentioned in the discussion section can be more easily understood. In addition, in the discussion section of the study "Investigation of the Associations of Novel Inflammatory Biomarkers—Systemic Inflammatory Index (SII) and Systemic Inflammatory Response Index (SIRI)—With the Severity of Coronary Artery Disease and Acute Coronary Syndrome Occurrence", which you offer for our review, you can also find similar information.
Reviewer 2 Report
Comments and Suggestions for Authors
In this manuscript, the authors reported that 37% of the patients were found to have a positive Fractional Flow Reserve (FFR), while 63% had a negative FFR. Significant differences were observed in neutrophil count, monocyte count, Systemic Inflammation Response Index (SIRI), total cholesterol, and the amount of adenosine used between the groups. A SIRI value of 1.16 was 77% sensitive and 55% specific for FFR positivity. Multivariate logistic regression analysis identified SIRI as an independent predictor of FFR positivity. This study indicates that high values of SIRI may serve as a new biomarker for predicting FFR positivity
However, considering many similar reports as indicated below, there is little novelty for this study.
[1] He T, Luo Y, Wan J, Hou L, Su K, Zhao J, Li Y. Analysis of the correlation between the Systemic Inflammatory Response Index and the severity of coronary vasculopathy. Biomol Biomed. 2024 Oct 17;24(6):1726-1734.
[2] Guo J, Huang Y, Pang L, Zhou Y, Yuan J, Zhou B, Fu M. Association of systemic inflammatory response index with ST segment elevation myocardial infarction and degree of coronary stenosis: a cross-sectional study. BMC Cardiovasc Disord. 2024 Feb 9;24(1):98.
[3] Dziedzic EA, GÄ…sior JS, Tuzimek A, Paleczny J, Junka A, DÄ…browski M, Jankowski P. Investigation of the Associations of Novel Inflammatory Biomarkers-Systemic Inflammatory Index (SII) and Systemic Inflammatory Response Index (SIRI)-With the Severity of Coronary Artery Disease and Acute Coronary Syndrome Occurrence. Int J Mol Sci. 2022 Aug 23;23(17):9553.
Additionally, most references are old and should be updated.
Comments on the Quality of English LanguageThe quality of English language is better. The references should be formatted through.
Author Response
Comments 1 However, considering many similar reports as indicated below, there is little novelty for this study.
Response 1:Dear reviewer, first of all, thank you for taking the time to evaluate our work. We partiallly agree with you. In one of the formentioned studies the authors studied the relationship between Gensini score and SIRI. In other two studies the study groups were acute coronary syndrome patients and the degree of stenosis were calculated by visual analyses. However we performed FFR all the patients that shows the functional features of stenosis,which is more accurete and objective. Therefore, it brings a new perspective that is different from the mentioned studies.
Round 2
Reviewer 1 Report
Comments and Suggestions for Authors
Once again I congratulate the authors for their work!
At this moment I have the following remarks.
1. In the materials and methods: please add the syntagma „all the patients that meet the inclusion criteria”, the exact period for your study (for example: 1 January 2012 - 31 December 2022), and the number of all patients treated in these two Centers. Again, and again, a flowchart will be magnificent. Move here, and elaborate as a flowchart Figure 1.
Add in all figures and tables Group 1 and Group 2.
For the previous 4 comment - if you choose to equalise your 2 Groups at this moment, it will be a source of doubt, so I think that is not suitable. Still, please mention this in the Introduction, in the material and method part, and in the Study limitations (these sentences are from your response, so please adapt them for the part/parts that suit you „Moderate stenosis is not common in angiograms. Whether these stenoses are functionally significant is evaluated by FFR examination. For this reason, unfortunately, our number of patients is low. If only our number of patients had been larger, we could have obtained better results”.)
2. In Results:
- in Table 2 please move the gender line after the age line, and add a body index line (BMI, if you have this possibility, if is not possible please mention that in the limitation part, and this is because there are studies that discuss „the obesity paradox”), and also the presence of the metabolic syndrome (also for its paradoxical comportment). For DM please mention if it is just type 2 DM, and if is not only type 2 mention the percentage for type 2 DM (as you mentioned in the previous response, add DM2 or type 2 DM, and explain in the abbreviation part). Also for an easier following of this table, please separate data from this table in different/appropriate tables.
3. Study limitation: please add the ideas mentioned above and those mentioned by you in the response to comment 5.
Author Response
Comment 1 In the materials and methods, please add the syntagma „all the patients that meet the inclusion criteria,” the exact period for your study (for example 1 January 2012 - 31 December 2022), and the number of all patients treated in these two Centers. Again and again, a flowchart will be magnificent. Move here, and elaborate as a flowchart Figure 1.
Add in all figures and tables Group 1 and Group 2.
For the previous four comment - if you choose to equalise your 2 Groups at this moment, it will be a source of doubt, so I think that is not suitable. Still, please mention this in the Introduction, in the material and method part, and in the Study limitations (these sentences are from your response, so please adapt them for the part/parts that suit you „Moderate stenosis is not common in angiograms. Whether these stenoses are functionally significant is evaluated by FFR examination. For this reason, unfortunately, our number of patients is low. If only our number of patients had been larger, we could have obtained better results”.)
Response 1:Dear Rewiever
Thank you again for sparing your valuable time and evaluating our work.
As you suggested, the date we recruited the patients was stated in the study.(page-2/line-65)The flowchart you specified was added as Figure 1 and the sentence "All the patients that meet the inclusion criteria" was stated in the material method section.(Page -2/Line-75-77)
Comment 2: In Results: - in Table 2 please move the gender line after the age line, and add a body index line (BMI, if you have this possibility, if is not possible please mention that in the limitation part, and this is because there are studies that discuss „the obesity paradox”), and also the presence of the metabolic syndrome (also for its paradoxical comportment). For DM please mention if it is just type 2 DM, and if is not only type 2 mention the percentage for type 2 DM (as you mentioned in the previous response, add DM2 or type 2 DM, and explain in the abbreviation part). Also for an easier following of this table, please separate data from this table in different/appropriate tables.
Response 2 As you mentioned, Table 2 was divided into two parts: basal characteristic features and biochemical features, Table 2 and Table-3.
The gender section was placed under the age section.
All patients were indicated as DM2 in the table because they had type 2 DM.
Since all patients included in the study did not have height-weight measurements, BMI calculation could not be made. It is also mentioned in the study limitations section due to the obesity paradox..(Page 8/ Line 257-259)
Comment 3:Study limitation: please add the ideas mentioned above and those mentioned by you in the response to comment 5.
Response 3:The limitations mentioned in comment 5 were mentioned in the study limitations section.(Page 8/ Line 254-257)
Reviewer 2 Report
Comments and Suggestions for Authors
This study suggests that high values of the Systemic Inflammation Response Index (SIRI) may serve as a new biomarker for predicting the positivity of Fractional Flow Reserve (FFR), hence is clinically significant and interesting.
Comments on the Quality of English LanguageThere are grammatical/spelling errors to be corrected.
The references should be formatted.
Author Response
Comment 1: There are grammatical/spelling errors to be corrected.
The references should be formatted.
Response 1:Dear Rewiever
Thank you again for sparing your valuable time and evaluating our work. We corrected grammatical/spelling errors and reformatted the references as you suggested.